# Paclitaxel-Loaded, Pegylated Carboxylic Graphene Oxide with High Colloidal Stability, Sustained, pH-Responsive Release and Strong Anticancer Effects on Lung Cancer A549 Cell Line

**DOI:** 10.3390/pharmaceutics16111452

**Published:** 2024-11-14

**Authors:** Athina Angelopoulou, Myria Papachristodoulou, Efstathia Voulgari, Andreas Mouikis, Panagiota Zygouri, Dimitrios P. Gournis, Konstantinos Avgoustakis

**Affiliations:** 1Department of Pharmacy, Medical School, University of Patras, 26504 Patras, Greeceefivoulgari48@gmail.com (E.V.); andreas27997@gmail.com (A.M.); 2Department of Materials Science and Engineering, University of Ioannina, 45110 Ioannina, Greece; pzygouri@gmail.com; 3School of Chemical and Environmental Engineering, Technical University of Crete, 73100 Chania, Greece; dgournis@tuc.gr; 4Institute of GeoEnergy, Foundation for Research and Technology-Hellas, 73100 Chania, Greece

**Keywords:** graphene oxide, paclitaxel, pegylation, colloidal stability, cytotoxicity, programmed cell death

## Abstract

**Background**: Graphene Oxide (GO) has shown great potential in biomedical applications for cancer therapeutics. The biosafety and stability issues of GO in biological media have been addressed by functionalization with polyethylene glycol (PEG). **Methods**: In this work, carboxylated, nanosized GO (nCGO) was evaluated as a potential carrier of paclitaxel (PCT). The effect of PEG characteristics on particle size and surface charge, colloidal stability, drug, and release, and the hemolytic potential of nCGO, was investigated. Optimum PEG-nCGO/PCT formulations based on the above properties were evaluated for their anticancer activity (cytotoxicity and apoptosis induction) in the A549 lung cancer cell line. **Results**: An increase in the length of linear PEG chains and the use of branched (4-arm) instead of linear PEG resulted in a decrease in hydrodynamic diameter and an increase in ζ potential of the pegylated nCGO particles. Pegylated nCGO exhibited high colloidal stability in phosphate-buffered saline and in cell culture media and low hemolytic effect, even at a relatively high concentration of 1 mg/mL. The molecular weight of PEG and branching adversely affected PCT loading. An increased rate of PCT release at an acidic pH of 6.0 compared to the physiological pH of 7.4 was observed with all types of pegylated nCGO/PCT. Pegylated nCGO exhibited lower cytotoxicity and apoptotic activity than non-pegylated nCGO. Cellular uptake of pegylated nCGO increased with incubation time with cells leading to increased cytotoxicity of PEG-nCGO/PCT with incubation time, which became higher than that of free PCT at 24 and 48 h of incubation. **Conclusions**: The increased biocompatibility of the pegylated nCGO and the enhanced anticancer activity of PEG-nCGO/PCT compared to free PCT are desirable properties with regard to the potential clinical application of PEG-nCGO/PCT as an anticancer nanomedicine.

## 1. Introduction

Graphene Oxide (GO) family nanomaterials, including GO, reduced GO (rGO), and carboxylated GO (CGO), have gained increased attention for biomedical applications, especially in the fields of drug delivery and theranostics [1,2,3,4]. The innate, hydrophobic properties of the pristine graphitic lattice of GO nanomaterials raise biosafety and poor biological stability issues, which limit their prospects for biomedical applications. To overcome such limitations, surface functionalization of GO nanomaterials is performed either by conjugation chemistry or through the exploitation of non-covalent interactions [3,5]. In order to increase biocompatibility and diminish safety concerns, GO nanomaterials have been functionalized with hydrophilic and biocompatible molecules, such as poly(ethylene glycol) (PEG) [6,7], dextran [8], bovine serum albumin (BSA) [7], heparin [9], chitosan [10], and polyacrylic acid [11]. GO disperses easily in water but precipitates in saline and physiological media. The colloidal stability of GO nanomaterials in physiological media is enhanced by the covalent attachment of hydrophilic polymers, such as PEG [12], poly(lactide)-poly(ethylene glycol) copolymers [13], dextran [14], pluronics [15], and tocopherol polyethylene glycol 1000 succinate (TPGS) [16].

Among the various conjugation processes, the most widely applied for the functionalization of GO nanomaterials has been the formation of amide linkages by EDC/NHS chemistry, the thiol linkages through dopamine, and the direct polymerization via linkages of initiators on the oxygen functional groups [3,5]. Such conjugation processes are used for the loading of drugs and small molecules, genes, antibodies, and aptamers on GO nanomaterials [4]. The non-covalent modifications include hydrophobic, electrostatic, hydrogen-bonding, and π–π stacking interactions [3,4,5]. Such interactions have made significant contributions to the adsorption of proteins and hydrophobic drugs, providing stabilization of the GO, rGO, and CGO and protection of the delivered molecules from proteolytic degradation. The gain obtained from covalent and non-covalent functionalization of GO, rGO, and CGO is enhanced biocompatibility, increased blood residence time, and elevated deep tumor accumulation [3,4,5]. Among the different covalent and non-covalent approaches, PEG and its derivatives have been extensively applied in GO nanomaterials functionalization in order to improve GO stability in biorelevant media and biocompatibility [17,18,19,20,21,22].

Recently, GO family nanomaterials have been intensively investigated as drug delivery systems (DDS) in combination with tumor-targeting peptides and folic acid (FA) molecules [17,18], metal and metal oxide nanoparticles [19,20], and aptamers [20,21], as well as in combination therapies, such as photothermal/photodynamic therapy and hyperthermia [22], to overcome drug resistance and improve anticancer efficacy. Li et al. [17] combined the effect of HN-1 (TSPLNIHNGQKL) tumor-targeting peptide and doxorubicin with nano-GO that was covalently functionalized with diamino-PEG. The drug doxorubicin (DOX) was loaded through π–π stacking and hydrogen bond interactions with the PEG-modified nano-GO. The formulation was evaluated for its HN-1 targeting effect against oral squamous cell carcinoma (OSCC) in CAL-27 and SCC-25 cell lines, showing significant internalization and increased anticancer effect (DOX cytotoxicity). In another study by Vinothini et al. [18], GO was grafted with methyl acrylate (MA) via in situ atom transfer radical polymerization, in order to create biodegradable GO surfaces for the conjugation of FA and the loading of PCT through π–π stacking interactions (GO-g-MA/FA). The targeting ability and cytotoxic effect of the formulations were examined in human MDA-MB-231 breast cancer cells. The in vivo evaluation of the PCT-loaded GO-g-MA/FA formulations in DMBA-induced breast cancer rats revealed their antitumor efficacy through the inhibition of tumor size and the regulation of mitochondrial mediate apoptotic cascade. The combination of DOX-loaded CGO with gold nanoparticles (GNP) was studied by Samadian et al. [19] for the development of a theranostic nanomedicine system. The CGO was functionalized with PEG-amine and thiol-terminated FA for the conjugation of GNP. The final system expressed pH-sensitive DOX release and elevated anticancer activity against MCF-7 human breast cancer cells. Hussien et al. [20] investigated the anticancer effect of superparamagnetic GO (MGO) formulations conjugated with MUC1 targeting aptamer and further loaded with PCT. The MGO formulations showed elevated biocompatibility in L-929 human normal fibroblasts. Moreover, the final formulations showed increased binding affinity to MCF-7 cancer cells and significant cytotoxicity. The effect of aptamer on the targeted delivery of CGO was examined by Yaghoubi et al. [10] in a human gastric adenocarcinoma AGS cell line. The CGO was functionalized with AS1411 aptamer for the co-delivery of curcumin and DOX. The successful targeting effect of the aptamer resulted in elevated cytotoxicity and the regulation of gene expression for cell cycle arrest (G1 phase) and apoptosis. GO was also investigated in thermally responsive formulations by Jedrzejczak-Silicka et al. [22]. The GO surface was functionalized with hydroxycamptothecin (GO-HCPT), and hyperthermia was induced via the application of a rotating magnetic field (RMF). The GO-HCPT nanomaterials exhibited high cytotoxicity against MCF-7 cancer cells and regulated mitochondrial metabolism, resulting in enhanced antitumor activity.

Among the various functionalizations used, PEG modification of nanoparticulate drug carriers is the most frequently applied, as pegylation provides for long circulation properties following intravenous administration. Furthermore, PEG has many desirable attributes, including the availability of various end-functional groups, which provide flexibility in coupling to nanoparticles with diverse chemical structures, the availability in linear or branched forms, and the availability in a large range of molecular weights. The above PEG attributes provide flexibility in the design of anticancer drug delivery and cancer theranostics [23,24,25,26]. Thus, the pegylation strategy was adopted in our study with the aim of developing stable in vivo and biocompatible PCT nanocarriers based on carboxylated graphene oxide. The characteristics of PEG, such as MW and linear or branched structure, can affect the surface properties and protein adsorption (opsonization of nanoparticles) [23,25]. Thus, in our study, the functionalization of CGO with PEG-NH_2_ polymers that vary in molecular weight (low and high MW) and polymer structure (linear and 4-arm) was investigated. The CGO served as an excellent surface for the covalent modification with PEG-NH_2_ and the efficient paclitaxel (PCT) loading, mainly through π–π stacking interactions, and CGO has been shown promising drug delivery results in previous studies [21,27,28]. The optimum PCT-loaded CGO formulation, based on physicochemical and drug release properties, was investigated for the in vitro anticancer efficacy in the A549 human lung cancer cell line.

## 2. Materials and Methods

### 2.1. Materials

Graphite flakes (purum, powder ≤ 0.2 mm), potassium chloride (purum > 98.0%), nitric acid (HNO_3_ 65 wt/wt%), and sulfuric acid (H_2_SO_4_ 95–97 wt/wt%) were obtained from Fluka. Monomethoxy poly(ethylene glycol)-amine (mPEG-NH_2_ of 2 kDa, 10 kDa, and 20 kDa), 4-armed poly(ethylene glycol)-amine (4-arm PEG-NH_2_ of 10 kDa), N-ethyl-N′-(3-dimethyl aminopropyl) carbodiimide hydrochloride (EDC-HCl, >98%, MW: 191.7 g/mol), N-hydroxysuccinimide (NHS, >98%, MW: 115.09 g/mol), triethylamine (Et_3_N, (C_2_H_5_)_3_N, >99.5%), and fluorescein isothiocyanate (FITC, >90% HPLC) were obtained from Sigma-Aldrich. Paclitaxel was purchased from LC Laboratories, Woburn, MA, USA. All other chemicals and solvents were of analytical grade.

### 2.2. Synthesis of Carboxylated Graphene Oxide

The oxidation of graphite for the synthesis of graphene oxide (GO) and further preparation of carboxylated graphene oxide (CGO) was followed and characterized as reported previously in Spyrou et al. [29] and Zygouri et al. [30]. Details for the successful synthesis and characterization of GO and CGO are reported in Appendix A.

### 2.3. Nano-Carboxylated Graphene Oxide (nCGO) Particles

#### 2.3.1. Preparation of the CGO Nanoparticles

Nanoparticles of CGO (nCGO) were prepared in aqueous suspensions. An initial concentration of 3 mg CGO was dispersed in 5 mL D-H_2_O through bath sonication (50 kHz, 350 W) for 2 h, followed by probe sonication (50/60 kHz, 130 W) in an ice-bath for an additional 3 h. The obtained CGO aqueous dispersion was centrifuged at 12,000× *g* for 30 min and filtered through Millipore filters of 0.2 μm diameter to provide nCGO particles. The nCGO yield measured by freeze-drying samples of the synthesized material and weighing the solid residues was 54% wt.

#### 2.3.2. Preparation of the Pegylated nCGO-PEG Nanoparticles

Amino-terminated poly(ethylene glycol) (PEG-NH_2_) polymers of varied molecular weights were utilized for the preparation of surface pegylated nCGO dispersions. EDC/NHS chemistry was used for the amide conjugation of PEG-NH_2_ with the activated free carboxyl groups of the nCGO, as in previous studies [31]. Thus, equimolar concentrations of 0.002 mmol mPEG-NH_2_ (2 kDa, 10 kDa, and 20 kDa) and 4-arm PEG-NH_2_ (10 kDa) were dissolved in 500 μL D-H_2_O and added dropwise in the nCGO dispersion under bath sonication for 5 min. Then, equimolar concentrations of 0.015 mmol EDC-HCl/NHS dissolved in 200 μL D-H_2_O were added dropwise. The pH of the reaction solution was adjusted to 8 with Et_3_N, so that the amino groups remain deprotonated and capable of reacting with the carboxylic acid groups. The reaction mixture was bath-sonicated in an ice-bath for 30 min and then gently stirred for 24 h at ambient temperature in the dark. The reaction product was purified by extended dialysis against water in the dark. Dialysis membranes of appropriate MWCO (14 kDa and 50 kDa) were used, depending on the MW of the PEG-NH_2_ polymers. Then, the purified product was centrifuged at 12,000× *g* for 30 min and filtered through Millipore filters of 1.2 μm diameter. The yield of the nCGO-PEG conjugates was in the range of 14–32% wt, depending on the MW of the mPEG-NH_2_ polymers.

#### 2.3.3. Fluorescent Labeling of the nCGO-PEG Nanoparticles

The nCGO-PEG(10 kDa) particles were labeled with FITC, which was conjugated through EDC chemistry. For the reaction process, 0.5 mg/mL nCGO-PEG was used and 0.5 mg EDC was added dropwise. The reaction mixture was sonicated in an ice-bath for 1 h for the activation of the carboxylic acid groups. Then, 100 μL FITC from a stock solution (1 mg/mL in D-H_2_O) was added dropwise and the pH was adjusted to 8 with Et_3_N. Following this, the mixture was gently stirred at ambient temperature for 24 h in the dark. The final product was purified by extended dialysis against D-H_2_O. The purified nano-CGO-PEG/FITC was stored in the fridge (4 °C) until further use.

#### 2.3.4. Characterization

The composition of the nCGO-PEG particles was examined by Thermal gravimetric analysis (TGA) on a TA Instrument Q500 series Thermogravimetric Analyzer (Waters, New Castle, IN, USA) at a heating rate of 10 °C/min from room temperature to 600 °C in N_2_. The nCGO-PEG conjugates were characterized by Fourier transform infrared spectra (FTIR) obtained from 600 to 4000 cm^−1^ using a DigiLab Excalibur series FTS 3000 spectrometer (Labexchange, Burladingen, Germany) equipped with an attenuated total reflectance (ATR) and a class II laser. The morphology of the nCGO-PEG particles was characterized by transmission electron microscopy (TEM) using a JEOL JEM-2100 microscope (JEOL, Peabody, MA, USA) at an acceleration potential of 200 kV, equipped with a GATAN camera Erlangshen ES500W, model 782 (JEOL, Peabody, MA, USA). Specimens for TEM were prepared by spreading an aqueous solution of 0.1 mg/mL concentration onto a carbon-coated Cooper MS200 grid (Polyscienses, Europe, Hirschberg an der Bergstrasse, Germany), which was air-dried before observation.

The colloidal stability of the nCGO-PEG particles was assessed by storing samples of the particles at room temperature (20 °C) for up to 4 weeks and by incubating the samples in mildly agitated PBS and RPMI 1640 cell culture medium (50 %*v*/*v* dilution) at 37 °C for 5, 24, and 48 h. The ζ-potential and average hydrodynamic size were recorded in a ZetaSizer Nano series Nano-ZS (Malvern Instruments Ltd., Malvern, UK) equipped with a He-Ne Laser beam at a wavelength of 633 nm. A fixed backscattering angle of 173° was used.

#### 2.3.5. Biocompatibility Assay

The biocompatibility of the nCGO and nCGO-PEG particles was assessed by hemolysis assay, as previously described [32]. Blood samples from healthy donors (obtained from the University Hospital of Patras, Patras, Greece) were collected in heparin tubes and centrifuged at 1000× *g* for 5 min for plasma separation. The red blood cells (RBC, erythrocytes) were washed thrice with normal saline and a ratio of 1:20 pure RBC was prepared at a final volume of 2 mL. The nCGO and nCGO-PEG particles were prepared in PBS for concentrations in the range of 0.5 μg/mL to 1000 μg/mL. Then, equal quantities of RBC and particles were mixed with a rotor shaker, incubated at 37 °C for 4 h, and were gently agitated every 30 min. Next, the samples were centrifuged at 1000× *g* for 5 min. The supernatant containing the hemoglobin was transferred in 96-well plates and the absorbance was measured at 570 nm using an MPR-700 Plate Reader, Biotech Engineering Management Co., Ltd. (San Francisco, CA, USA). For positive and negative controls, RBCs were exposed to 10% Triton X-100 solution (+, positive, 100% hemolysis) and PBS (−, negative, 0% hemolysis) at identical conditions. The test was repeated thrice, and the percent of hemolysis was calculated from the following equation:(1)$$Hemolysis\;\%=\frac{A_{nCGO}-A_{negative}}{A_{positive}-A_{negative}}\times 100$$
where A_nCGO_, A_positive_, and A_negative_ are the absorbances of hemoglobin contained in the supernatant of the nCGO and nCGO-PEG particles, in the negative control and in the positive control, respectively.

### 2.4. Paclitaxel Loading and Release from nCGO-PEG Particles

For the drug loading process, a paclitaxel (PCT) solution (1 mg/250 μL) methanol was added dropwise in 500 μL of nCGO-PEG dispersions. The reaction mixture was bath sonicated for 15 min and gently agitated for 24 h, in the dark. Then, the reaction mixture was dialyzed against excess D-H_2_O for 24 h in a dialysis membrane (MWCO 12 kDa) in order to remove the organic solvent and non-entrapped PCT. Drug loading was measured by UV–Vis spectroscopy, in a UV-1800 Shimadzu spectrophotometer (SHIMADZU scientific instruments, Long Beach, CA, USA). The nCGO-PEG/PCT particles were analyzed at 227 nm [33], with the absorbance of blank (no drug) nCGO-PEG particles being extracted upon measurement. PCT quantification was based on a calibration curve with R^2^ = 0.9976 and limit of quantification 0.01 μg/mL, with the linear part of the standard curve being from 0.01 to 100 μg/mL. The dispersion solution for PCT quantification was D-H_2_O:MeOH = 60:40 in %*v*/*v*. The PCT loading capacity was calculated according to the following formula:(2)$$LC\;\%=\frac{W}{W_{C}+W}\times 100$$
where W and W_C_ were the amount of loaded PCT according to the UV–Vis spectroscopy and the number of nCGO-PEG particles, respectively.

The release profile of PCT from the nCGO-PEG/PCT particles was obtained under sink conditions in 1xPBS buffer pH 7.4 and 6.0. In brief, nCGO-PEG/PCT samples were enclosed in dialysis membranes (MWCO 12 kDa) and transferred to vials containing 20 mL PBS. The vials were put in a mildly agitated water bath (37 °C). At predetermined time intervals (30 min, 1, 2, 4, 6, 12, 24, 48 h), the release medium was completely removed and replaced with fresh PBS at 37 °C. The release medium was treated with 2 mL of DCM (dichloromethane) for the extraction of the released PCT. The extraction process was repeated in triplicate for each sample. Then, DCM containing the released PCT was collected, allowed to evaporate at room temperature in a fume hood, and the solid residue was dissolved in 1 mL of D-H_2_O:MeOH = 60:40 (%*v*/*v*). The obtained solutions were assayed for PCT by UV–Vis spectroscopy at 227 nm, as described above.

### 2.5. Cellular Evaluation of the nCGO/PEG Particles

The human A549 (ATCC) lung adenocarcinoma epithelial cells were cultured under standard conditions [32]. The anticancer activity of nCGO-PEG/PCT particles against the A549 cells was assessed using the propidium iodide (PI) fluorescence method [31]. The cells were seeded in 24-well plates (5 × 10^4^ cells per well) and proliferated for 24 h (37 °C in a humidified atm. with 5% CO_2_). Then, the cells were treated with PCT and nCGO-PEG/PCT particles at indicated concentrations (0, 1, 5, 10, 17, 25 μg/mL of drug) for varied time periods of 5, 24, and 48 h. The blank nCGO-PEG (no drug) and nCGO (no PEG, no drug) particles were also evaluated as controls at similar concentrations to the drug-loaded ones. The cells were then washed with PBS, harvested (0.25 %*w*/*v* trypsin), and transferred to FACS tubes for centrifugation (1600 rpm for 5 min). Then, the cells were stained with PBS containing 5 μL PI (1 mg/mL PI stock solution) for 1 min. The cell death distribution was determined via flow cytometry analysis (for PI fluorescence with λ_exc._ = 488 nm, λ_em._ = 620 nm) in a FACS Calibur, Coulter Epics XL-MCL apparatus (system II software). Unlabeled cells were used as a negative control to calculate the background fluorescence of the cells. Data analysis was performed with the WinMDI cytometry analysis software (version 2.8).

For the cellular uptake experiments, nCGO and nCGO-PEG particles were labeled with FITC. The A549 cells were treated with 10 μg/mL nCGO-PEG/FITC and nCGO-FITC samples for varied time periods (1, 5, 24, and 48 h). The cells were grown in monolayers and then harvested and washed with PBS. The cellular uptake was quantitatively evaluated by flow cytometry (FITC fluorescence λ_exc._ = 488 nm, λ_em._ = 543 nm) in the FACS Calibur analyzer. The background fluorescence of untreated cells was used as a negative control. The visualization of the cellular uptake was performed following the PI post-fixation staining method proposed by Hezel et al. [34] The PI staining method is normally used for the quantitative assessment of cell death. In the post-fixation PI method, cellular fixation with paraformaldehyde and disruption of the cellular membrane with Triton X are performed prior to PI staining. This way, PI is allowed to diffuse and interact with nuclear and cytoplasmic nucleic acid. Thus, intact non-degenerating cells may be visualized. In the post-fixation staining method, cells were grown in coverslips and treated with 10 μg/mL nCGO-PEG/FITC particles for 24 h. Then, the cells were washed with PBS, and fixed with 4 %*v*/*v* PFA (paraformaldehyde) for 15 min directly on the cover slips. Following, the cells were washed, and cellular membrane disruption was promoted by 0.1 %*v*/*v* Triton X 100 (Sigma Aldrich, Merck, St. Louis, MO, USA) for 10 min. Finally, the cells were washed with PBS and treated with PI for 15 min. The final specimens were imaged in a Leica Microsystems DMLB I/2001 microscope (Microscope Central, Feasterville-Trevose, PA, USA) equipped with a Leica fluorescence source and a Leica DC 300 camera.

In the Cell apoptosis study, the cells were treated with PCT and nCGO-PEG/PCT at 10 μg/mL for 1, 5, 24, and 48 h. The nCGO-PEG and nCGO particles were used as the control. After treatment, the cells were harvested and washed with PBS. Then, cellular apoptosis was determined with the FITC Annexin V Apoptosis staining method according to a standard protocol [32]. In brief, cells were washed with PBS, harvested, and centrifuged in FACS tubes (1600 rpm, 5 min). Then, cells were washed and re-suspended to 1xAnnexin V binding buffer and centrifuged (1600 rpm, 5 min). Cells were then re-suspended to 100 μL 1xAnnexin V binding buffer with 5 μL FITC Annexin V solution (in the dark, 15 min). Finally, the cells were washed with 1xAnnexin V binding buffer (Sigma Aldrich, Merck). Apoptosis was analyzed in the FACS Calibur analyzer (Annexin V fluorescence λ_exc._ = 495 nm, λ_em._ = 519 nm). The background fluorescence of untreated cells was used as a negative control.

### 2.6. Statistical Analysis

For the statistical analysis of experimental data, appropriate statistical methods (Student’s *t*-test and one-way ANOVA) were applied using the IBM SPSS Statistics 25 software.

## 3. Results

### 3.1. Characterization

The successful functionalization of nCGO with PEG-amine was shown with FTIR, ^1^H NMR, and TGA. In the FTIR spectra of the nCGO-PEG particles (Figure 1A, Table 1), all the characteristic peaks of nCGO were observed, including the broad band around 3400 cm^−1^ attributed to the stretching vibrations of -OH groups, the peak at 1630 cm^−1^ ascribed to the stretching vibrations of -C=O, the peak at 1386 cm^−1^ denoting the deformation vibrations of the C-OH groups, and the band at 1061 cm^−1^ ascribed to the stretching vibrations of the C-O-C. Moreover, the main peaks of PEG polymers were recognized at 2883 cm^−1^ attributed to the stretching vibration of methylene (-CH_2_-) groups, and the peak at 1100 cm^−1^ was ascribed to the stretching vibrations of C-O-C groups. The effective conjugation of PEG on nCGO was verified by the following bands in the spectrum of nCGO-PEG, which do not exist in the spectrum of nCGO: (a) the band at 1625 cm^−1^ attributed to the stretching vibration of the -C=O- group of amide-type I, (b) the band at 1592 cm^−1^ ascribed to the stretching vibration of -NH of amide-type II, and (c) the band at 2883 cm^−1^ attributed to the stretching vibration of methylene (-CH_2_-) groups of PEG [35,36,37,38,39]. The proton NMR spectra of the nCGO, PEG-amine, and nCGO-PEG were recorded in D_2_O (Appendix A). The nCGO and nCGO-PEG spectra exhibited a peak at 2.21 ppm attributed to the nCGO methylene (-CH_2_-) protons accompanied by varied weak peaks in the range 1.0–2.0 ppm of the aliphatic area attributed to the CH-CH protons of GO. They also had a low peak at 8.5 ppm, which is attributed to the aromatic protons of the graphite lattice [17,40,41]. The peak at 3.69 ppm in the spectrum of PEG-amine and nCGO-PEG is ascribed to the methylene (-CH_2_-) protons of PEG.

The TGA thermograms of the nCGO-PEG particles (Figure 1 and Appendix A) exhibited mass loss profiles that combined the characteristics of the nCGO matrix and PEG polymers and were dependent on the MW of the polymers. The nCGO exhibited an initial stage of mass loss of about 10% at 100−150 °C, which can be attributed to the evaporation of adsorbed water. A mass loss of only 8% was observed at 150–400 °C due to the high thermal stability of the -COOH complexes. A final third stage of mass loss of around 10% was observed at 400–600 °C, which was due to the decomposition of the carbon skeleton. For PEG polymers, complete decomposition was observed at 350–420 °C, regardless of the MW of the polymer [13,19]. In order to determine the exact composition of the nCGO-PEG conjugates (proportion of CGO and PEG-NH_2_) from the thermograms (Table 2), the equation (1 − M) n = W [13] was used, where M is the polymer percent weight of the composite, n is the percent residual weight of nCGO, and W is the percent residual weight of the composite.

The average hydrodynamic size and ζ-potential of the nCGO and nCGO-PEG particles are presented in Table 2. According to DLS, the average size (hydrodynamic diameter) of the nCGO was around 80 nm, while the average size of the nCGO-PEG particles ranged between 100 and 180 nm. The increased size of the nCGO-PEG conjugates was also verified by TEM (Figure 1 and Appendix A) and was probably due to the presence of the hydrated PEG molecules on the particles and possibly to the slight nanosheets aggregation caused by the reaction between activated carboxyl groups (-COOH) in neighboring nanosheets forming anhydrides [36]. The conjugation of PEG appeared to have an impact on the average size of the nCGO-PEG particles that depended on the MW and structure of PEG. An increase in the MW of PEG caused a decrease in the size of the particles (Table 2), presumably due to the more effective steric stabilization of the conjugates by the high MW PEG molecules. Also, the four-arm PEG appeared to be a more effective steric stabilizer than the one-arm PEG, resulting in lower particle size for the nCGO-PEG particles functionalized with the four-arm PEG polymer (Table 2).

The ζ-potential of the nCGO was highly negative (−50 mV) due to the presence of –COOH groups on the lattice. After PEG conjugation, the nCGO-PEG particles exhibited an increase in ζ-potential (lower negative values, Table 2) that depended on the MW of the PEG polymers, ranging from −20 mV to −35 mV. The increased ζ potential of the pegylated nCGO further confirmed the successful PEG conjugation on the graphene oxide surface [21].

### 3.2. Colloidal Stability of nCGO-PEG

The results of the colloidal stability study of the nCGO and nCGO-PEG particles are shown in Figure 2 and Appendix A. The size and ζ potential of the nCGO and nCGO-PEG particles remained essentially constant with storage at ambient temperature for a period of four weeks, indicating good physical stability (Appendix A). The only exception was the nCGO-PEG(2 kDa) particles, which showed an increase in size after a week’s time storage, assuming a value around 300 nm, which, however, remained essentially constant for the remaining storage time (Figure 2A,B). All nCGO-PEG particles, with the exception of nCGO-PEG(2 kDa) particles, were stable in PBS and only small changes in size and ζ-potential of the particles were observed during the 48 h incubation period (Figure 2C,D). The size and ζ-potential of the nCGO-PEG particles tended to increase with incubation time in RPMI 1640 cells culture medium; however, no aggregation or sedimentation was observed. Again, and in accordance with the storage stability results (Figure 2A), the nCGO-PEG(2 kDa) particles were the less stable particles and tended to aggregate in RPMI 1640 with incubation time, reaching a size of around 500 nm at 48 h (Figure 2C).

### 3.3. Hemolysis Assay Results

The hemolytic potential of the nCGO and nCGO-PEG particles was determined as an indication of particle biocompatibility. Nanoparticles should not cause hemolysis if they are intended to be applied as drug delivery systems entering the bloodstream. The particles were evaluated for their hemolytic activity by the disruption of the erythrocytes membrane in human red blood cells (Figure 3). The hemolysis caused by nCGO-PEG was very low, lower than 5%, even at the high concentration of 1 mg/mL. The pristine nCGO particles caused higher hemolysis than the nCGO-PEG particles, reaching a hemolysis value of around 15% at the concentration of 1 mg/mL [42].

### 3.4. Loading and Release Profiles from nCGO-PEG/PCT Particles

The main interactions facilitating PCT loading on graphene and graphene oxide sheets are via π–π stacking and hydrophobic interactions, mainly promoted by the increased binding energy of PCT with both graphene and graphene oxide. Thus, PCT adsorption on graphene and graphene oxide is favorable through the three benzene rings that provide PCT with a strong hydrophobic character in aqueous media [43]. In this study, we applied pegylation on CGO nanosheets, and it was found that the MW and structure of the conjugated PEG-NH_2_ had an impact on the loading capacity of PCT (Table 2). Among linear PEG-NH_2_ conjugates, the loading capacity on PCT increased as the MW of the polymer decreased; thus, nCGO-PEG(2 kDa) showed the highest loading of PCT at around 67 %wt, while nCGO-PEG(20 kDa) the lowest near 38 %wt. The 4-arm PEG(10 kDa)-NH_2_ exhibited a PCT loading capacity of 33 %wt, lower than the 52 %wt loading of the linear PEG(10 kDa)-NH_2_.

The nCGO-PEG/PCT particles exhibited sustained slow PCT release at pH 7.4 with no burst effect (Figure 4A). Less than 40% of the drug was released within the evaluated time period of 48 h. The lowest release rate was shown by the nCGO-PEG(2 kDa) around 30%, while the high MW polymer particles showed similar release profiles. At pH 6.0 (Figure 4B), a higher PCT release was observed for all particles compared to the release at pH 7.4. The released PCT was almost 90% at 48 h for the nCGO-PEG(10 kDa) particles, while the low MW nCGO-PEG(2 kDa) had a release near 60%. In the case of nCGO-4armPEG(10 kDa) particles, a burst effect occurred at pH 6.0, with a 50% release in the first hour.

### 3.5. Results from the Cell Studies

For the in vitro cellular evaluation, the nCGO-PEG(10 kDa)/PCT formulation was selected, which had optimum physicochemical characteristics. Moreover, in a recent study by Khramtsov et al. [44], it was found that the type of PEG coating (branched or linear) affected the internalization of GO nanoparticles by cells, and branched PEG-modified GO particles were less prompt to be internalized by cells than the linear PEG-modified ones, mainly due to the protein adsorption on the particles. Thus, the nCGO-PEG(10 kDa)/PCT particles were opted for further study compared to the nCGO-4 arm PEG(10 kDa)/PCT particles. Endocytosis is an important feature of graphene and graphene-based nanomedicine systems to effectively transport drug molecules within cellular compartments via receptor-mediated and non-specific mechanisms [1,45].

The cytotoxicity of nCGO-PEG(10 kDa) against the A549 cell line was investigated at 5, 24, and 48 h of incubation. The blank (non-loaded with PCT) nCGO-PEG particles had lower cytotoxicity at all concentrations and incubation times than the blank nCGO particles (Figure 5), indicating an increased biocompatibility of the pegylated graphene oxide particles. Free PCT and nCGO-PEG(10 kDa)/PCT particles exhibited comparable cytotoxicity at 5 h of incubation that was close to 25% at 25 μg/mL of PCT (Appendix A). At longer incubation time periods, the nCGO-PEG/PCT exhibited significantly (*p* < 0.01) higher cytotoxicity compared to free PCT, reaching the concentration of 25 μg/mL of PCT with almost 60% and 75% cell destruction at 24 and 48 h, respectively.

To further evaluate the in vitro anticancer activity of the nCGO-PEG/PCT particles, their effect on cellular apoptosis was investigated. For this study, the nCGO (no PEG, no PCT) and the nCGO-PEG(10 kDa) (no PCT) were evaluated as control samples, and the effects of PCT and nCGO-PEG/PCT were compared with regard to the apoptotic damage induced in A549 cells. Figure 5F shows that treatment of the cells with 10 μg/mL PCT and nCGO-PEG(10 kDa)/PCT induced apoptosis in a time-dependent mode. The apoptosis level induced at 48 h in the A549 cells by the nCGO, nCGO-PEG, PCT, and nCGO-PEG/PCT particles was around 11 ± 0.77%, 8 ± 0.72%, 18 ± 1.56%, and 27 ± 1.72%, respectively. Moreover, the apoptosis analysis revealed that the PCT and nCGO-PEG/PCT reached a maximum of induced apoptosis at around 24 h. The results showed that at all time-points the cell apoptosis was higher after nCGO-PEG/PCT treatment.

The internalization of nCGO and nCGO-PEG(10 kDa) was evaluated quantitatively by FACS after labeling the particles with FITC. Significant uptake with an almost linear increase with time was observed for the nCGO-PEG particles incubated with the A549 cells for 5, 24, and 48 h (Figure 5C). The uptake of nCGO-PEG particles was significantly (*p* < 0.01) higher than that of nCGO particles at all times tested (Figure 5C). The post-fixation PI staining method was used to visualize the cellular uptake of the nCGO-PEG particles [21]. Under the experimental conditions used, PI can enter dead A549 cells (Figure 5D) and colocalize with FITC-labeled particles inside the cells to provide yellow-colored areas (Figure 5E). Thus, the fluorescence microscopy images demonstrated the uptake of nCGO-PEG particles by the A549 cancer cells.

## 4. Discussion

Increased research interest has been devoted to the functionalization of GO family nanomaterials (including GO, rGO, and CGO) in order to optimize their therapeutic applications [44,45,46,47,48,49]. The advances offered by the modification of GO nanomaterials with PEG-based polymers have increased the prospects of their potential application in drug delivery and cancer therapeutics [49]. The synergistic effects of GO nanomaterials with chemotherapeutic agents, photodynamic/photothermal therapy, and sonotherapy represent a recent cutting-edge field [50,51,52,53]. Shi et al. [50] reported the functionalization of rGO with PEG(5 kDa) for the effective conjugation of the TRC105 anti-CD105 antibody. A bifunctional succinimidyl carboxymethyl-PEG(5 kDa)-maleimide (SCM-PEG-mal) polymer was used for the stabilization of nano-rGO at a size range of 20–80 nm. Then, the PET isotope p-SCN-Bn-NOTA was effectively conjugated via a terminal amine linkage and the TRC105 antibody via Traut’s reagent. The functionalized rGO expressed elevated photothermal properties and acted as a theranostic agent, by combining PET imaging and tumor vasculature CD105 targeting. Moreover, Guo et al. [51] studied the modification of GO with PEG(2 kDa)-bisNH_2_ polymer through EDC/NHS chemistry for the conjugation of oxidized sodium alginate and the loading of paclitaxel. The system showed pH/thermal-responsive drug release and was evaluated against paclitaxel-resistant HGC-27 gastric cancer cells. Under NIR-irradiation, the GO system promoted an elevated rate of apoptosis and mitochondrial damage of the HGC-27 cells, due to the combined photothermal/photodynamic effect of the GO lattice and anticancer effect of paclitaxel. In another study by Lee et al. [52], 4-arm PEG(2 kDa)-NH_2_ was used for the modification of GO and Graphene nanoribbons (GNR) for the delivery of the sonosensitizer chlorin e6 (Ce6). GNR-4arm PEG exhibited enhanced colloidal stability in cell culture media for up to 48 h, while GO-4arm PEG was stable for up to 24 h. In comparison, GNR-4arm PEG expressed superior inhibition effects on tumor spheroid adhesion than GO-4arm PEG. Also, the GNR-4arm PEG system was evaluated on the inhibition and sonodynamic effect against ovarian cancer spheroids, resulting in reduced adhesion of the spheroids and reduced metastatic potential. Clearly, the binding affinity and biocompatibility of GO family nanomaterials can be significantly enhanced by PEG polymers’ modification, reducing the tendency to aggregation, enhancing cytocompatibility effects, and regulating hemocompatibility [42,52]. In a recent study by Khramtsov et al. [44], it was found that the polymeric chain of PEG (linear or branched) affected serum protein adsorption on GO-PEG, due to diverse steric hindrance effects generated by the different polymeric chains. The uptake of the GO-PEG nanoparticles from monocytes that were isolated from peripheral blood cells was also affected, resulting in reduced internalization of the nanoparticles functionalized with the branched PEG polymer.

In our study, we investigated the effect of PEG MW and structure (linear or 4-arm branched PEG 10 kDa) on the basic physicochemical characteristics (Table 2), colloidal stability (Figure 2), hemocompatibility (Figure 3), and release properties (Figure 4), of nCGO-PEG/PCT particles. Since the modification of GO with linear or branched PEG has not been connected with differences in the observed cytotoxicity [44,53], the optimum nCGO-PEG/PCT particles were evaluated in cellular studies involving A549 lung adenocarcinoma cells (Figure 5).

The experimental process that was followed for the preparation of nCGO particles involved continuous steps of sonication/centrifugation/filtration, for the most stable aqueous nCGO suspension to be isolated. This process resulted in nCGO particles with a desired size of 76 nm and ζ-potential close to −48 mV, which were further used for the conjugation of the various PEG-NH_2_ polymers. Pegylation increased the hydrodynamic size and ζ-potential of nCGO particles (Table 2). The increase in hydrodynamic size is due to the presence of the hydrated layer of PEG on the nCGO surface, and the decrease in the negative value of ζ-potential is due to the neutralizing effect of the electrically neutral PEG chains. An increase in the molecular weight of the (linear) PEG from 2 to 20 kDa decreased the size and tended to increase the ζ-potential of the nCGO-PEG particles (Table 2), indicating a more effective stabilizing and charge screening effect of the high MW PEG polymers. PEG provides steric stabilization of nCGO particles, and the higher MW PEG or the branched PEG generates a more efficient barrier to particle aggregation, leading to lower particle sizes. Also, branched PEG resulted in nCGO-PEG particles having a lower size and higher ζ-potential compared to the linear PEG of the same PEG chain length (10 kDa) (Table 2), due to the more effective coverage of graphene oxide surface by the branched polymer. Similar effects were observed by Liu et al. [54], who applied linear and branched PEG(40 kDa) to stabilize nanoemulsions.

All nCGO-PEG particle types produced in this work exhibited good colloidal stability on storage in PBS and cell culture medium, with the exception of the nCGO-PEG functionalized with the relatively low MW PEG of 2 kDa (Figure 2). Apparently, the relatively low molecular weight PEG chains of 2 kDa were unable to prevent a small degree of aggregation of the nCGO-PEG(2 kDa) particles. Pegylation decreased significantly the hemolytic potential of nCGO-PEG particles (Figure 3). All nCGO-PEG particles caused less than 5% hemolysis, even at a relatively high concentration of 1 mg/mL. This result, taken together with the lack of cytotoxicity of the nCGO-PEG particles (Figure 5), suggests the biocompatibility of the nCGO-PEG nanocarrier. The increased hemolytic potential of pristine nCGO (15% at 1 mg/mL) is probably due to the electrostatic interactions of the RBCs with the plentiful oxygen groups of the nCGO surface. Moreover, the hydrophobic interactions between the pristine nCGO lattice and the phospholipids on the RBCs membrane may lead to the damage and disruption of the cellular membrane [42]. Pegylation increased the biocompatibility of graphene oxide particles by reducing the available oxygen molecules on the matrix and by offering a protective surface layer that conceals the electrostatic interactions of oxygen molecules and the hydrophobic interactions of nCGO lattice with the RBCs.

For the delivery of PCT, non-covalent loading through physical adsorption was performed for all the nCGO-PEG particles, exploiting the π–π stacking interactions between the nCGO lattice and PCTs’ aromatic rings [4,5]. An increase in PEG MW caused a decrease in PCT loading (Table 2). PEG chains probably generate a hindrance to the efficient approach of PCT on the nCGO surface, thus reducing drug loading capacity, and the longer the chains of PEG, the more pronounced this effect is. In line with our results and reasoning, Zhao and Liu [55] reported that the doxorubicin loading capacity of pegylated GO was lower than that of the GO, probably because the larger functional PEG brushes on the GO reduced the interaction between the drug molecules and the nanocarrier, and Khramtsov et al. [44] reported that branched PEG more efficiently shields GO surfaces than linear PEG, which was attributed to the steric hindrance generated by the multiple PEG chains, and reduced more efficiently protein adsorption on GO. In agreement with a more efficient shielding of GO surface by the branched (multi-armed) PEG, we observed a lower PCT loading capacity in nCGO-PEG(4-arm 10 kDa) compared to nCGO-PEG(10 kDa) (Table 2). Support for the more efficient surface shielding by multi-armed PEG provides the relatively low PCT loading (11.2 %wt) reported by Xu et al. [45] for graphene oxide particles functionalized with a 6-armed poly(ethylene glycol). Zhuang et al. [53] conjugated carboxylic acid-modified PCT on graphene oxide 4-arm PEG-folic acid particles and succeeded an 18.7% drug loading; thus, the covalent PCT conjugation did not provide any improvement with regard to PCT loading on CGO.

At pH 7.4, a slow drug release was observed from all types of nCGO-PEG/PCT particles, with less than 40% PCT being released in 48 h (Figure 4A). Low PCT leakage from the nanoparticles at physiological pH is necessary to avoid off-target accumulation of drugs before the nanoparticles reach the tumor area. Increased PCT release rate was observed at acidic pH 6.0 for all nCGO-PEG/PCT particles (Figure 4B). The hydrogen-bonding interaction between PCT and GO is strongest at close to neutral conditions (pH 7.4) compared to the acidic conditions, and this may be the reason for the increased release rate observed at acidic pH. In line with this reasoning, Yang et al. [56] reported higher doxorubicin release from GO at acidic and basic pH, which was attributed to the partial dissociation of hydrogen-bonding interaction under acid and basic pH conditions, and Hussien et al. [20] observed a higher PCT release rate from magnetic GO nanocarriers at acidic pHs compared to pH 7.4. The nCGO-PEG(2 kDa) particles exhibited the lowest PCT release at both pH environments (Figure 4). The lower PCT release rate from the nCGO-PEG(2 kDa) particles at both pHs studied can probably be attributed to the reduced surface coverage provided by the relatively low MW polymeric chains of 2 kDa PEG. This allowed PCT to create strong π–π stacking and hydrophobic interactions with the graphite lattice. In the case of nCGO-4armPEG(10 kDa) particles, a burst effect occurred at pH 6.0. An increased drug release rate at acidic pH is a desirable attribute for anticancer nanomedicines since it can provide a more selective drug delivery in the acidic environment of tumors.

Among the various particles investigated, the nCGO-PEG(10 kDa)/PCT exhibited an optimal combination of properties including average size (<150 nm), colloidal stability, hemocompatibility, PCT loading, and release at pH 7.4 (<40%) and pH 6.0 (~90%). Thus, the nCGO-PEG(10 kDa)/PCT was selected for cellular studies. At 5 h, the nCGO-PEG/PCT and the free PCT exhibited comparable cytotoxicity values against the A549 cell line, while at 24 h and 48 h, the nCGO-PEG(10 kDa)/PCT significantly surpassed the anticancer activity of free PCT (Figure 5A,B). Specifically, the IC50 value at 48 h of nCGO-PEG(10 kDa)/PCT was close to 5 μg/mL of the drug, while free PCT showed IC50 near 25 μg/mL. The higher cytotoxicity of the particle-entrapped PCT compared to free PCT at higher incubation times may be attributed to the increasing cellular uptake with time of nanoparticles (Figure 5C). The cytotoxicity of nCGO-PEG(10 kDa)/PCT reached the concentration of 25 μg/mL on a PCT basis of 60% and 75% cell destruction at 24 and 48 h, respectively. A little lower cytotoxicity against the A549 cell line was reported for nano-GO particles stabilized with PLA-PEG copolymers [13]. The effect of PCT-loaded, 6-armed PEG-functionalized graphene oxide particles on the viability of A549 cells was investigated by Xu et al. [46], who reported a significantly increased cell death close to 70% at 1.29 mg/L particle concentration. In a more recent study by Lin et al. [47], graphene oxide was reduced with *Euphorbia milii* leaf extract and then was loaded with PCT. The reduced graphene oxide/PCT complexes showed a dose-dependent reduction in A549 cell viability that reached 10% at the concentration of 500 μg/mL after a 24-h incubation period. The increased cytotoxicity of the nCGO-PEG/PCT is particularly important for the potential application of nCGO-PEG as a PCT delivery system, since, if confirmed in vivo, it would provide increased therapeutic efficacy compared to the free drug. The increased therapeutic efficacy will allow for the reduction in dosing frequency or dose size in the clinical setting, both of which will lead to decreased side effects and increased tolerance of the pharmacotherapy from the patients.

To further investigate the anticancer ability of nCGO-PEG/PCT particles in relation to free PCT, the apoptotic effect against A549 cells was examined (Figure 5F). The results showed an increased presence of apoptotic cells at 24 and 48 h by the nCGO-PEG/PCT particles, reaching a maximum at 24 h post-incubation. In a recent study by Zhao et al. [34], the antitumor effect and the mechanism of induced apoptosis of paclitaxel and paclitaxel nanoparticles on A549 cells were investigated. The evaluation concluded that paclitaxel NPs significantly inhibited the G2 phase of the cell cycle and increased programmed cell death. The outmost of the effect on G2 phase and cellular apoptosis was observed at 15 h after light onset (HALO). The nCGO-PEG particles caused less than 10% of programmed cell death at 48 h, while a slightly increased apoptosis rate was induced by the nCGO control samples. The effect of PEG modification on cellular uptake was also evaluated (Figure 5C). The nCGO control samples exhibited a significantly reduced uptake in comparison to the nCGO-PEG at 24 and 48 h, supporting the elevated tumor cell accumulation ability provided by the functionalization of nCGO nanosheets with PEG [4,5].

Pegylation appears to have a positive effect on nCGO biocompatibility as evidenced by the hemolysis and cytotoxicity results (Figure 3 and Figure 5); however, a more thorough investigation, involving in vivo experiments, is required in order to fully assess the safety of nCGO-PEG nanocarriers for drug delivery applications.

## 5. Conclusions

Based on the results obtained in this study, the structural characteristics (molecular weight and branching) of PEG significantly affected the physicochemical and drug release properties of the pegylated CGO nanoparticles. The nCGO-PEG/PCT nanoparticles exhibited sustained, pH-responsive drug release properties, having a significantly higher drug release at pH 6.0 compared to the physiological pH 7.4. No or low drug release at physiological pH is a desirable attribute for anticancer drug nanocarriers for limiting off-target drug release and side effects of the drug to healthy tissues. The significantly improved biocompatibility characteristics of the nCGO-PEG nanocarriers were evidenced by the reduced hemolytic potential and decreased cytotoxicity of the pegylated particles compared to CGO controls. The pegylated CGO nanoparticles exhibited high cellular uptake by the A 549 cancer cell line at the longer incubation times tested, which led to strong anticancer activity upon incubation of the nCGO-PEG/PCT nanoparticles with the A549 cells, stronger than the free drug at the longer incubation times. The obtained results justify further investigation into the potential of nCGO-PEG/PCT for the efficient destruction of tumors in vivo.

## Figures and Tables

**Figure 1 pharmaceutics-16-01452-f001:**
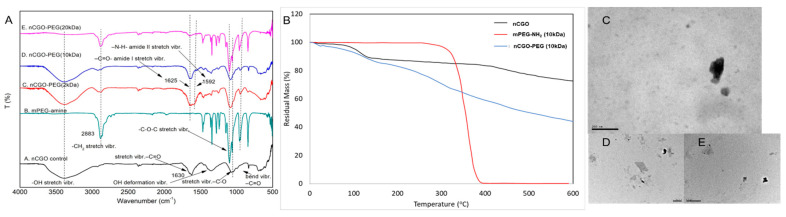
Characterization of nCGO-PEG particles: (**A**) FTIR spectra of nCGO (black) mPEG-NH_2_ (amine) polymer (light blue), and nCGO-PEG particles of varied MW (2, 10, 20 kDa) (red, blue magenta); (**B**) thermograms of nCGO, mPEG(10 kDa)-NH_2_, and nCGO-PEG(10 kDa) up to 600 °C; (**C**) SEM micrographs of nCGO-PEG(10 kDa) at scale bar of 200 nm (**C**), 50 nm (**D**), and 100 nm (**E**).

**Figure 2 pharmaceutics-16-01452-f002:**
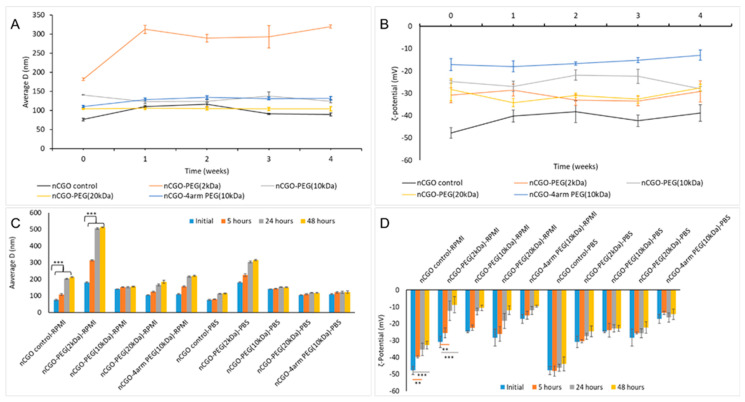
Colloidal stability of the nCGO and nCGO-PEG particles exhibiting (**A**) average size distribution by DLS and (**B**) distribution of ζ-potential for a period of 4 weeks. The stability of nCGO and nCGO-PEG particles in RPMI and PBS media as presented by (**C**) average size and (**D**) ζ-potential at 5, 54, and 48 h. The statistical significance is ** *p* < 0.001, *** *p* < 0.0001.

**Figure 3 pharmaceutics-16-01452-f003:**
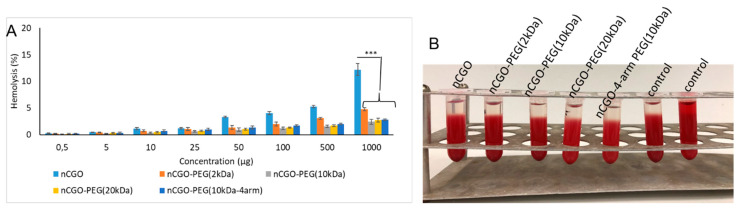
(**A**) Hemolysis observed at varied concentrations of the nCGO and nCGO-PEG particles. (**B**) Representative hemolysis photographs at a particle concentration of 25 μg/mL and with the control (positive, negative) samples. The statistical significance is *** *p* < 0.0001.

**Figure 4 pharmaceutics-16-01452-f004:**
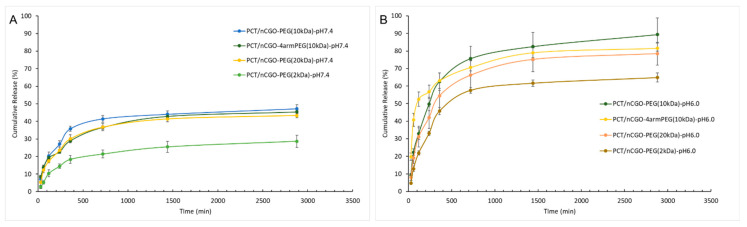
Paclitaxel release profile from the PCT/nCGO-PEG particles in PBS buffer with pH (**A**) 7.4 and (**B**) 6.0.

**Figure 5 pharmaceutics-16-01452-f005:**
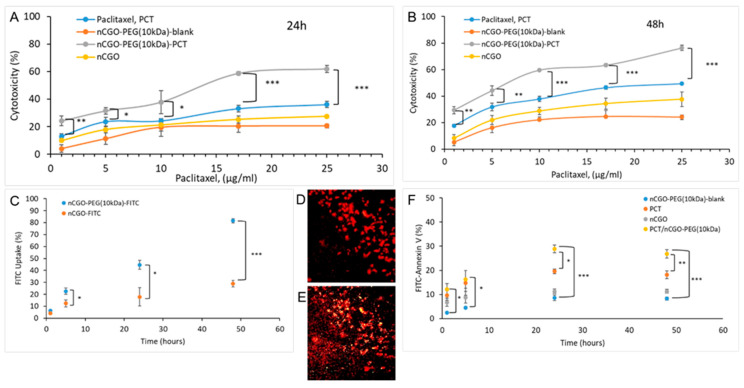
Evaluation of the nCGO-PEG(10 kDa) particles against lung adenocarcinoma A549 cell line for the induced anticancer effect (cytotoxicity) at (**A**) 24 h and (**B**) 48 h. Internalization of FITC-labeled nCGO-PEG(10 kDa) in comparison with nCGO-particles (**C**) and Fluorescence microscopy by PI post-fixation staining method of A549 cellular nuclei (**D**) and cells treated with FITC-labeled nCGO-PEG(10 kDa) particles (**E**). Evaluation on programmed cell death of A549 cells by apoptosis assay induced by nCGO (grey circles), nCGO-PEG(10 kDa) blank (blue circles), PCT (orange circle), and nCGO-PEG(10 kDa)/PCT loaded (yellow circle) particles (**F**). The statistical significance is * *p* < 0.01, ** *p* < 0.001, *** *p* < 0.0001.

**Table 1 pharmaceutics-16-01452-t001:** Characteristic FTIR peaks.

Sample	IR Absorption Bands (cm^−1^)	Description
GO-COOH	1061	Stretching vibration of C-O-C bonds
	1386	Deformation vibration of C-OH bond
	1634	stretching vibrations of -C=O
	3400	Hydroxyl (-OH) stretching vibration bands (broad peak)
nGO-PEG	1625	Stretching vibration of -C=O- amide I
	1592	Stretching vibration of -N-H- amide II
	2883	Stretch vibrations of methylene -CH_2_
PEG-amine	1100	stretch vibrations of -C-O-C bond
	2883	Stretch vibrations of methylene -CH_2_

**Table 2 pharmaceutics-16-01452-t002:** Characteristics of the nCGO and nCGO-PEG particles of varied PEG MW (2, 10, 20 kDa) and structure (linear, 4-arm).

Sample	Yield (%)	CGO Weight Ratio (%)	PEG-NH_2_ Weight Ratio (%)	PCT Loading (%)	Average Size (nm)	PdI	ζ-Potential (mV)
nCGO	54.89 ± 6.85	-	-	-	76.10 ± 3.49	0.354 ± 0.036	−47.82 ± 2.36
CGO-PEG(2 kDa)/PCT	26.85 ± 0.90	23.31	76.69	67.64 ± 1.32	181.4 ± 3.40	0.431 ± 0.051	−30.84 ± 3.36
CGO-PEG(10 kDa)/PCT	14.63 ± 2.69	60.84	39.2	52.43 ± 2.38	140.5 ± 1.02	0.311 ± 0.089	−24.78 ± 0.61
CGO-PEG(20 kDa)/PCT	32.38 ± 0.85	20.40	79.6	38.88 ± 2.42	104.2 ± 1.16	0.276 ± 0.012	−28.35 ± 5.04
CGO-PEG(4-arm 10 kDa)/PCT	23.61 ± 1.51	21.95	78	33.66 ± 1.74	110.3 ± 2.37	0.209 ± 0.023	−17.14 ± 2.65

## Data Availability

Data is contained within the article or Appendix A.

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
