# Peer review of "Paclitaxel-Loaded, Pegylated Carboxylic Graphene Oxide with High Colloidal Stability, Sustained, pH-Responsive Release and Strong Anticancer Effects on Lung Cancer A549 Cell Line"

_pharmaceutics, 2024, doi:10.3390/pharmaceutics16111452_

Round 1
Reviewer 1 Report
Comments and Suggestions for Authors
In this paper "Pegylated carboxylic graphene oxide for controlled paclitaxel delivery to tumor cells: in vitro evaluation and cell studies" the authors reported about the syhthesis of Pegylated carboxylic graphene oxide and its application for controlled paclitaxel 2 delivery to tumor cells.
However, I think the paper is still showing some weakness in various perspectives. Therefore, I recommend the authors to consider the following comments to address.
The title is very general so need to revise some catchy title.
The abstract should be rewritten well by incorporating some advantages how these finding based on synthesized material can be impactful to clinical application.
Need to mention in introduction for choosing pegylation of the material and its potential properties.
Why authors mention the synthesis in both files.
Based on the role of pegylation, it would be valuable to mention any future prospects for in vivo biocompatibility assessments if applicable.
Need to discuss a clearer mechanistic insight, how the length of PEG chain length affects the π-π (pi-pi) stacking interactions with paclitaxel (PCT).
Need to mention about the pH sensitivity arises from PEG itself or from its interaction with PCT.
Need to highlight the significance of the results of cytotoxicity in terms of dosing frequency or reduced side effects.
Conclusion should be revised by adding all the revised results.
Comments on the Quality of English LanguageEnglish language need to improve throughout the manuscript.
Author Response
Authors wish to thank the Reviewers for their constructive comments and suggestions, which helped authors to improve the scientific value of the paper.
Reviewer 1
In this paper "Pegylated carboxylic graphene oxide for controlled paclitaxel delivery to tumor cells: in vitro evaluation and cell studies" the authors reported about the syhthesis of Pegylated carboxylic graphene oxide and its application for controlled paclitaxel 2 delivery to tumor cells.
However, I think the paper is still showing some weakness in various perspectives. Therefore, I recommend the authors to consider the following comments to address.
- The title is very general so need to revise some catchy title.
Response:
The title has been changed to “Paclitaxel-loaded, pegylated carboxylic graphene oxide with high colloidal stability, sustained, pH-responsive release and strong anticancer effects on lung cancer A549 cell line” which we believe better represents the work done.
- The abstract should be rewritten well by incorporating some advantages how these finding based on synthesized material can be impactful to clinical application.
Response:
The abstract was rewritten in order to present the offered advantages of our nanoparticles for potential clinical applications.
- Need to mention in introduction for choosing pegylation of the material and its potential properties.
Response:
The significance of GO pegylation has been added in the introduction (page 3 lines 110-119, yellow highlighted).
- Why authors mention the synthesis in both files.
Response:
The synthesis and characterization of graphene oxide (GO) and carboxylated graphene oxide (CGO) have been studied and published in references [29, 30] of our colleagues. Thus, the detailed synthetic process was added in the Supporting Information in order to avoid repetition.
- Based on the role of pegylation, it would be valuable to mention any future prospects for in vivo biocompatibility assessments if applicable.
Response:
A paragraph has been added that connects the role of PEGylation assessed in our research, with future prospects for in vivo biocompatibility assessments (page 15, lines 652-655).
- Need to discuss a clearer mechanistic insight, how the length of PEG chain length affects the π-π (pi-pi) stacking interactions with paclitaxel (PCT).
Response:
The length of PEG chains does not interfere with the π-π (pi-pi) stacking interactions of GO with PCT. We believe, based on previous reports, that PEG chains generate a hindrance on the efficient approach of PCT on GO surface, thus reducing drug loading capacity, and the longer the chains of PEG the more pronounced this effect is. In line with our results and reasoning, Zhao and Liu [1] reported that the doxorubicin loading capacity of pegylated GO was lower than that of the GO, probably because the larger functional PEG brushes on the GO reduced the interaction between the drug molecules and the nanocarrier, and Khramtsov et al. [2] reported that branched PEG more efficiently shield GO surfaces than linear PEG, which can be explained by steric hindrance generated by multiple PEG chains, and reduced more efficiently protein adsorption on GO.
[1] Xubo Zhao and Peng Liu, Biocompatible graphene oxide as a folate receptor-targeting drug delivery system for the controlled release of anti-cancer drugs, RSC Adv., 2014, 4, 24232-24239, DOI: 10.1039/C4RA02466D.
[2] Khramtsov et al., Interaction of Graphene Oxide Modified with Linear and Branched PEG with Monocytes Isolated from Human Blood. Nanomaterials. 2022, 12, 126. https://doi.org/10.3390/ nano12010126.
The effect of the length of PEG chains on PCT loading is discussed in page 13 (lines 574-585) of the revised manuscript (yellow highlighted).
- Need to mention about the pH sensitivity arises from PEG itself or from its interaction with PCT.
Response:
Hydrogen-bonding interaction between PCT and GO is strongest at close to neutral conditions (pH 7.4) compared to the acidic conditions and this may be the reason for the increased release rate observed at acidic pH. In line with this reasoning, Yang et al. [3] reported higher doxorubicin release from GO at acidic and basic pH, which was attributed to the partial dissociation of hydrogen-bonding interaction under acid and basic pH conditions.
[3] Xiaoying Yang, Xiaoyan Zhang, Zunfeng Liu, Yanfeng Ma, Yi Huang, and Yongsheng Chen, High-Efficiency Loading and Controlled Release of Doxorubicin Hydrochloride on Graphene Oxide, The Journal of Physical Chemistry C 2008 112 (45), 17554-17558, DOI: 10.1021/jp806751k.
The effect of pH on the release profile of PCT is discussed in page 14 lines 595-603 (yellow highlighted).
- Need to highlight the significance of the results of cytotoxicity in terms of dosing frequency or reduced side effects.
Response:
The significance of the cytotoxicity results in terms of dosing frequency or reduced side effects are discussed in page 14 (lines 631-636) of the revised manuscript (yellow highlighted).
- Conclusion should be revised by adding all the revised results.
Response:
The Conclusions section has been rewritten to accommodate revisions made.
- English language need to improve throughout the manuscript.
A special effort has been devoted in improving English of the manuscript.

Reviewer 2 Report
Comments and Suggestions for Authors
This is a well-prepared paper that can be accepted for publication after minor revisions. The following issues should be addressed:
-
It would be helpful to divide subsection 2.3 into smaller sections, for example: preparation of nanocarboxylated graphene oxide particles; fabrication of PEGylated carboxylic graphene oxide particles...
-
Please present the FTIR analysis results in a table format.
-
If possible, please include TGA curves for all studied samples of PEGylated carboxylic graphene oxide, and provide TGA data as differential thermogravimetry.
-
The DLS results should be explained in more detail. Please include a discussion on the impact of PEG molecular weight and structure on the size and ζ-potential of the particles. It is unclear why PEGylated nanoparticles with higher molecular weight exhibit a smaller average size.
Author Response
Authors wish to thank the Reviewers for their constructive comments and suggestions, which helped authors to improve the scientific value of the paper.
Reviewer 2
This is a well-prepared paper that can be accepted for publication after minor revisions. The following issues should be addressed:
- It would be helpful to divide subsection 2.3 into smaller sections, for example: preparation of nanocarboxylated graphene oxide particles; fabrication of PEGylated carboxylic graphene oxide particles...
Response:
The following subsections have been added in the revised manuscript. 2.3.1 Preparation of the CGO nanoparticles, 2.3.2 Preparation of the Pegylated nCGO-PEG nanoparticles, 2.3.3 Fluorescent labelling of the nCGO-PEG nanoparticles, 2.3.4 Characterization, 2.3.5 Biocompatibility Assay.
- Please present the FTIR analysis results in a table format.
Response:
The FTIR results have been presented in a Table format in the revised manuscript (Table 1).
- If possible, please include TGA curves for all studied samples of PEGylated carboxylic graphene oxide, and provide TGA data as differential thermogravimetry.
Response:
The TGA thermograms of all samples are presented in detail in the Supporting Information part. The acquisition of TGA was performed in a collaborating laboratory and it is not possible to provide differential thermogravimetric results since they were not recorded. The TGA was used supplementary for evaluating the extent of GO pegylation.
- The DLS results should be explained in more detail. Please include a discussion on the impact of PEG molecular weight and structure on the size and ζ-potential of the particles. It is unclear why PEGylated nanoparticles with higher molecular weight exhibit a smaller average size.
Response:
A discussion on the impact of PEG molecular weight and structure on the size and ζ-potential of the particles has been added in the discussion section of the revised manuscript (page 13, lines 541-554, green highlighted). PEG provides steric stabilization of GO particles and the higher MW PEG or the branched PEG generate a more efficient barrier to particles aggregation, leading to lower particle sizes.
